

# Exogenous application of nitric oxide donors regulates short-term flooding stress in soybean

Muhammad Aaqil Khan[1], Abdul Latif Khan[2], Qari Muhammad Imran[1], Sajjad Asaf[2], Sang-Uk Lee[1], Byung-Wook Yun[1], Muhammad Hamayun[3], Tae-Han Kim[4] and In-Jung Lee[1]

[1] School of Applied Biosciences, Kyungpook National University, Degue, South Korea
[2] Natural and Medical Sciences Research Center University of Nizwa, Nizwa, Oman
[3] Department of Botany, Abdul Wali Khan University, Mardan, Pakistan
[4] School of Agricultural Civil & Bio-industrial Machinery Engineering, Kyungpook National University, Daegu, South Korea

## ABSTRACT

Short-term water submergence to soybean (*Glycine max* L.) create hypoxic conditions hindering plant growth and productivity. Nitric oxide (NO) is considered a stress-signalling and stress-evading molecule, however, little is known about its role during flooding stress. We elucidated the role of sodium nitroprusside (SNP) and S-nitroso L-cysteine (CySNO) as NO donor in modulation of flooding stress-related bio-chemicals and genetic determinants of associated nitrosative stress to Daewon and Pungsannamul soybean cultivars after 3 h and 6 h of flooding stress. The results showed that exogenous SNP and CySNO induced glutathione activity and reduced the resulting superoxide anion contents during short-term flooding in Pungsannamul soybean. The exo-SNP and CySNO triggered the endogenous *S*-nitrosothiols, and resulted in elevated abscisic acid (ABA) contents in both soybean cultivars overtime. To know the role of ABA and NO related genes in short-term flooding stress, the mRNA expression of *S-nitrosoglutathione reductase* (*GSNOR1*), *NO overproducer1* (*NOX1*) and *nitrate reductase* (*NR*), *Timing of CAB expression1* (*TOC1*), and *ABA-receptor* (*ABAR*) were assessed. The transcripts accumulation of *GSNOR1, NOX1*, and *NR* being responsible for NO homeostasis, were significantly high in response to early or later phases of flooding stress. *ABAR* and *TOC1* showed a decrease in transcript accumulation in both soybean plants treated with exogenous SNP and CySNO. The exo- SNP and CySNO could impinge a variety of biochemical and transcriptional programs that can mitigate the negative effects of short-term flooding stress in soybean.

# INTRODUCTION

Flooding is an abiotic stress which adversely affects the growth and productivity of different plant species all over the world (*Jackson & Colmer (2005)*. It is estimated that global warming might impact 70% of the world population associated with agriculture, and that the losses incurred through floods will increase up to five folds (*Alfieri et al., 2016*).

Corresponding author
In-Jung Lee, ijlee@knu.ac.kr

Flooding results in number of crises including slow gas exchange and low light levels under water which are responsible for both carbon and energy deficit leading to plant death (*Sasidharan et al., 2018*). Submergence also changes the redox status of the cell and reports suggested that flooding may alter the concentrations of oxygen-derived free radicals such as reactive oxygen species (ROS), and nitric oxide (NO), (*Bailey-Serres & Voesenek, 2008*). Flooding stress is fatal to all higher plants including crops, however, the impact will be much higher in crops sensitive to flooding stress. One of the most sensitive crop to flooding is the economically important soybean, as millions of tons of soybean are lost due to flooding in different regions of South America (*Bailey-Serres, Lee & Brinton, 2012*). There is an estimated 25% reduction in soybean yield due to flood injuries in North America, Asia and other regions of the world where soybean is rotated with rice in paddy fields (*Mustafa & Komatsu, 2014*; *Tewari & Arora, 2016*). Flooding stress adversely affects the growth and productivity of soybean with an approximate 17%–43% reduction at the vegetative stage while 50%–56% at the reproductive stage (*Mustafa & Komatsu, 2014*; *Visser & Pierik, 2007*).

Flooding stress generates hypoxic conditions which lead to the unavailability of molecular oxygen to the non-photosynthetic tissues of the plants. The prolonged submergence results in the manifestation of stress ethylene, whereas, the excessive amount of ethylene inhibits root elongation and ultimately cause plant death (*Visser & Pierik, 2007*), which is often removed by the aerenchyma; however, this depends on the level, intensity, and duration of flooding stress. Additionally, this further signals a plethora of physiological networks associated with plant growth and flooding stress. It involves activating different endogenous phytohormones (abscisic acid-ABA, gibberellins-GA, and auxin) and antioxidant enzymes (peroxidase, and catalases). Abscisic acid (ABA) have been reported for its potential role in flooding stress conditions and also known as stress phytohormone because of its response and distinct role in plant adaptation during abiotic stresses. Endogenous ABA levels increase rapidly, activating specific signaling pathways and modifying gene expression levels in response to abiotic stress (*O'Brien & Benkova, 2013*).

Several studies have been reported changes in ABA concentrations in different plants, including alfalfa (*Castonguay, Nadeau & Simard, 1993*), tobacco (*Hurng et al., 1994*), pea (*Jackson, Young & Hall, 1988*), tomato (*Else et al., 1995*), gerbera daisy (*Olivella et al., 2000*); Citrus (*Arbona & Gómez-Cadenas, 2008*; *Rodriguez-Gamir et al., 2011*); and apple (*Bai et al., 2011*) under flooding stress. The ABA response to flooding may differ and depend on the plant species and duration of flooding. Other studies reported that flooding stress have associated ABA accumulation with an increase in reactive oxygen species (ROS), in root parts of *Glycine max* L (*Vantoai & Bolles, 1991*), and *Triticum aestivum* L. (*Biemelt et al., 2000*) and leaf part of *Zea mays* L. (*Yan et al., 1996*). A time-course study in *Arabidopsis thaliana* L. showed that ABA increased with flooding stress, triggering stomatal closure and changes in hydrogen peroxide ($H_2O_2$) followed by an increase in antioxidant enzyme activities (*Liu et al., 2012*). The role of ABA is pivotally important in transcriptional regulation of protein-encoding genes and has been estimated up to 10% (*Nemhauser, Hong & Chory, 2006*).

Plants respond to environmental stresses through complex mechanisms involving regulatory genes that mediate signal transduction during stress. These unfavorable conditions results in the production of important molecules called reactive nitrogen intermediates (RNIs) (*Burniston & Wilson, 2008*), which regulate multiple physiological and biochemical processes in plants (*Durzan & Pedroso, 2002*; *Valderrama et al., 2007*). The primary source of production of these RNIs is NO, which has gained significant attention during the last couple of decades (*Delledonne et al., 1998*; *Durner, Wendehenne & Klessig, 1998*; *Yun et al., 2011*). After its recognition as a signaling molecule, NO has been reported to play a key role in plant resistance against abiotic and biotic stressors (*Yu et al., 2014*; *Yun et al., 2011*). The production of NO and the mechanism of its action have been well elucidated in animals (*Alderton, Cooper & Knowles, 2001*) but remain unclear in plants. There are two major pathways for NO biosynthesis reported in plants to date: the oxidative route involving NO synthesis from L-arginine (L- Arg) polyamines or hydroxyl amines, and the reductive route involving nitrate reductase (NR) enzyme (*Yu et al., 2014*).

Reactive nitrogen species (RNS), the NO derived molecules are relatively less known redox-active molecules that like ROS are produced after stress perception. The most studied among them is the group of S-nitrosothiols (SNOs) that is formed by the interaction of NO with intracellular sulfhydryl-containing molecules and are of great interest as they are more stable in solution than is NO. Therefore, they help in intracellular signaling, storage, and delivery of NO (*Leterrier et al., 2011*). One of the important SNOs is S-nitrosoglutathione (GSNO) which is formed by the S-nitrosylation reaction of NO with glutathione (GSH) and is NO mobile reservoir thereby having a strong impact on cellular redox status (*Shahzad et al., 2016*). GSNO reductase (GSNOR) is an enzyme that catalyses the NADH-dependent reduction of GSNO to GSSG and NH3 (*Jensen, Belka & Du Bois, 1998*). Therefore, GSNOR activity can regulate cellular NO homeostasis (*Leterrier et al., 2011*). *NOX1* (*nitrous oxide overexpressor* 1) also reportedly involved in NO metabolism (*Wang et al., 2017a*). Mutation in the gene resulted in increased accumulation of NO resulting in reduced root growth phenotype (*Wang et al., 2017a*). This suggests that *NOX1* positively regulates NO homeostasis. In addition, transcriptomic analysis has made it easy to show the global changes in gene expression in response to a particular stimulus. A number of potential candidate genes have been reported to regulate various biological processes in *Arabidopsis* by *Hussain et al. (2016)* and in *Trifolium repens* by *Zhang et al. (2018)*. However, detail investigation involving *in vitro* analyses are required to validate particular gene's function. There are other reports that described the negative role of NO through N-end rule pathway (i) Ac/N-end rule pathway, (ii) Arg/N-end rule pathway or phytoglobine. When plants were subjected to flooding stress like ethylene the concentration of NO also increase due to the restriction of gas diffusion. However, NO is highly reactive and react with oxygen to produce nitrite and NO$_3$ or with ROS to form additional reactive nitrogen species (*Delledonne et al., 2001*; *Chamizo-Ampudia et al., 2017*). In addition, (*Magalhaes, Monte & Durzan, 2000*; *Manac'h-Little, Igamberdiev & Hill, 2005*) reported that exo-NO application increases ethylene production in various plant species, possibly via enhanced ACO activity. NO role under hypoxic condition has also been described. Reports suggested that ERFVIIs are important transcriptional regulators of hypoxia responses of
hypoxia adaptive gene expression (*Gasch et al., 2016*; *Gibbs et al., 2011*; *Hinz et al., 2010*; *Licausi et al., 2011*). Studies have revealed that the presence of a characteristic N-terminal motif makes ERFVIIs direct targets of the N-end rule pathway of proteolysis, where, in the presence of both ambient NO and oxygen, these proteins are ubiquitinated and degraded in Arabidopsis (*Gibbs et al., 2011*; *Gibbs et al., 2014*; *Licausi et al., 2011*). When either the NO or oxygen concentration declines, ERFVIIs are no longer flagged for ubiquitination, accumulate, and activate downstream target genes.

Application of exogenous inducers to plants may help them evade unwanted environmental conditions; for example, Thiamine, a compound that alleviates oxidative stress during different abiotic stress conditions (*Tunc-Ozdemir et al., 2009*). Similarly, exogenous NO application to soybean plants subjected to flooding stress has been largely ignored in previous studies. NO has a well-established role in the key physiological processes relating to plant development and immunity (*Neill et al., 2008*; *Yu et al., 2014*; *Yun et al., 2016*). Its reactivity is also of crucial importance for its signaling functions, as NO is a highly reactive and toxic radical endogenously produced by plants in response to a variety of stressors (*Wang et al., 2013*). Additionally, NO plays a central role in plant growth and development (*Shi et al., 2007*; *Talukdar, 2013*; *Wang et al., 2017b*). Studies have shown that exogenous NO application mitigates the stress induced by salinity (*Shi et al., 2007*), high or low temperatures (*Hasanuzzaman, Hossain & Fujita, 2012*), heavy metals (*Talukdar, 2013*; *Wang et al., 2017b*), drought (*Jday et al., 2016*), and toxic chemicals (*Asgher et al., 2016*; *Khan et al., 2017*) in various crop plants. *Alderton, Cooper & Knowles (2001)* reported that SNP donated NO stimulates overall photosystem II electron transport rate.

Some recent studies have elucidated that exogenous NO application alleviates flooding stress. These studies demonstrated that the application of NO helps in plant fitness against hypoxia and waterlogging (*Dordas, 2015*; *Fan et al., 2017*; *Gupta, Lee & Ratcliffe, 2016*). The regulatory role of NO in biotic and abiotic stress conditions have been reviewed in detail (*Misra, Misra & Singh, 2011*). However, the effects of NO in regulating endogenous phytohormones and the implications of NO-induced redox imbalance in mitigating flooding stress have not been investigated fully. In the present study, we aimed to understand the interactive role of exogenous SNP and CySNO as NO donor at phytohormonal and molecular level of the plant under short-term flooding stress. Hence, a 3 h and 6 h flooding stress was given to soybean plants (Daewon; normal and Pungsannamul; sensitive) and the exogenous NO was applied to see the regulation of oxidative stress (superoxide anion, reduce glutathione), phytohormone (abscisic acid), molecular transcript (*NOX1*, *GSNOR1*, *NR*, *ABAR*, and *TOC1*) and total *S*-nitrosothiol during stress conditions. Furthermore, we studied the molecular characterization of NO and ABA related transcripts of potent genes in both soybean cultivars under short term flooding stress.

## MATERIALS AND METHODS

### Plant growth, flooding stress and exogenous NO application

Soybean seeds (*Glycine max* L.) of two varieties (Daewon; normal and Pungsannamul; sensitive) were collected from Kyungpook National University's Genetic Resource Centre,

Republic of Korea. The seeds were surface sterilized using sodium hypochlorite (2.5%) solution for 20 mint on a shaker at 120 rpm and washed twice in distilled water. The seeds were then subjected to a germination assay in plastic trays under greenhouse conditions with day/night cycle of 14 h at 28 °C/10 h at 25 °C and relative humidity of 60%–70% with natural light exposure. Uniform seedlings were selected randomly at leaf stage 1 (VI) and nine plants were planted in each plastic tray (41 cm × 24.5 cm) containing nutrient-rich soil (peat moss (10%–15%), perlite (35%–40%), coco peat (45%–50%), zeolite (6%–8%), $NH_4^+$~0.09 mg ● $g^{-1}$, $NO_3^-$~0.205 mg ● $g^{-1}$, $P_2O_5$~0.35 mg ● $g^{-1}$, and $K_2O$ ~0.1 mg ● $g^{-1}$). This makes a total of 45 plants for each treatment structure.

When the plants reached vegetative leaf stage 3 (V3), flooding stress was applied to maintain the water level up to 7 cm above the soil surface (Partial submergence). The experimental design (randomized block design) consisted of a control in which soybeans did not receive any treatment (Cont.), soybean plants subjected to flood stress only (CF), flood stress in addition to the application of 250 μM SNP (from stock solution of 100 mM SNP take 2.5 ml/L to make final concentration of 250 μM SNP), and flooding stress in addition to the application of exogenous 250 μM CySNO (from stock solution of 250 mM CySNO take 1 ml/L to make final concentration of 250 μM). The amount of NO donor (SNP and CySNO) were pre-dissolved in distilled water and applied on soybean plants compared to control treatments. After 3 and 6 h of flooding stress the plants were harvested, frozen in liquid nitrogen and stored at −70 °C for further analyses. Each treatment was comprised of five pots, each pot containing nine plants. Thus making a total of 45 plants to assess the effect of individual treatment. The biochemical, genes expression and phytohormonal analysis were performed in triplicate.

## ROS generation and quantification

For superoxide anion production analysis, the detailed method of (*De Sousa et al., 2017*; *Doke, 1983*; *Gajewska & Skłodowska, 2007*) were followed by measuring the reduction of exogenously supplied nitroblue tetrazolium (NBT). 1 g of fresh shoot plant powder sample was immersed in 0.01 M sodium phosphate buffer (pH 7.0) containing 0.05% (w/v) NBT and 10 mM sodium azide ($NaN_3$) and kept for 1 h at room temperature. Thereafter, 5 ml of the solution was transferred into a new test tube and heated for 15 min at 85 °C in a water bath. Immediately after heating, the solution was cooled down in ice and vacuum filtered. The absorbance of the sample was read at 580 nm on Shimadzu spectrometer (Shimadzu, Kyoto, Japan). Superoxide anion scavenging activity was calculated using the following equation: % Scavenging = ($A_{580}$ Control − $A_{580}$ Sample/$A_{580}$ Control) × 100. The experiment was repeated in triplicate.

To determine the reduction in glutathione content was estimated using the detailed procedure described by *Asaf et al. (2017)*, *Ellman (1959)* and *Khan et al. (2017)*. In brief, each sample (500 mg) were powdered in a chilled mortar and pestle with liquid nitrogen and was treated with 3 ml of 10% Trichloro Acetic Acid (TCA). The homogenate was centrifuged at 4 °C for 15 min at 10,000 × g. The resulting supernatant was then transferred into a new tube. One milliliter of the supernatant was combined with 0.5 ml μl of 5.5'-dithio-bis (2-nitrobenzoic acid) (75.3 mg in 30 ml of 100 mM phosphate buffer pH 6.8): 3 ml of 150

mM sodium phosphate buffer (NaH$_2$PO$_4$, pH 7.4) and were then incubated at 30 °C for 5 min. The absorbance of the solution was measured at 412 nm with the spectrophotometer (Shimadzu, Kyoto, Japan) and the GSH content was estimated using a standard curve. The experiment was repeated three time for each sample.

## Endogenous SNO quantification

SNO quantification was done as described earlier (*Islam et al., 2016*). Briefly, 100 mg plant tissue (shoot) was ground in liquid nitrogen to a fine powder using a mortar and pestle. The ground sample was then mixed with 1 ml extraction buffer (1X PBS, pH 5.4) and centrifuged at 13,000× g for 10 min. The supernatant was transferred to a fresh tube and again centrifuged at 15,000× g for 10 min. Proteins were quantified using the Bradford assay according to the manufacturer's standard protocol. A standard curve was constructed to measure protein concentration by measuring optical density (OD) of bovine serum albumin (BSA) standards at 595 nm wavelength using spectrophotometer. For SNO determination, 100 µl of the plant extracts were injected into the purge vessel of the NO Analyser (NOA 280; Sievers, Boulder, Co, USA) containing the reducing buffer (CuCl/cysteine with water); the peak values were recorded. The SNO contetns (pmol. µg −1 protein) in each samples were calculated using standard curve plotted against the OD595 values for each standard of CysNO. The standard curve was used to calculate total *S*-nitrosothiol levels in each sample.

## Endogenous ABA quantification

At harvest, soybean plants (shoot) were immediately frozen in liquid nitrogen and ground using a mortar and pestle. Endogenous ABA content was analysed following the modified protocol adopted from *Kamboj et al. (1999)* and *Qi et al. (1998)*. The powdered samples were treated with 30 ml of extraction solution containing 95% isopropanol, 5% glacial acetic acid, and 20 ng of [(±)–3,5,5,7,7,7–d$^6$]–ABA. The extracts were dried and methylated by adding diazomethane for GC/MS-SIM (6890 N network GC system, and 5973 network mass-selective detector; Agilent Technologies, Palo Alto, CA, USA) analysis. For quantification, the Lab-Base (Thermo Quest, Manchester, UK) data system software was used to monitor responses to ions of *m/e* 162 and 190 for Me-ABA and 166 and 194 for Me- [$^2$H$_6$]-ABA.

## RT-PCR analysis of RNA transcript during flooding

The protocol of *Chan et al. (2004)* was adopted with some modifications. By using TRIzol$^{TM}$ reagents the total RNAs were extracted from soybean leaves using the manufacturer standard protocol (Thermo Scientific; USA). For complementary DNA (cDNA) synthesis a total of 1 microgram RNA was used to reverse transcribed by using DiaStar$^{TM}$ RT kit (SolGent, Daejeon, South Korea). The synthesized cDNA (1 µl) was used as a template in quantitative real time PCR (qRT PCR) machine (Illumina, San Diego, CA, USA) using 2 × Real-time PCR Master Mix (including SYBR® Green I BioFACT$^{TM}$ Korea). A no template control containing PCR-grade water instead of template DNA was used as negative control. Tow-step PCR was performed for 40 cycles of amplification using selected primers and actin. The PCR conditions were, polymerease activation at 95 °C for
**Table 1  List of primers used for NO and ABA related genes through RT-PCR.**

| Gene name | Locus ID | Forward primer (5′-3′) | Reverse primer (5′-3′) |
|-----------|----------|------------------------|------------------------|
| TOC1-1 | NM_001248273 | TGGCAGCTTGGACTTGTAGA | ATTCACCCCTTGTTGAGCAC |
| GmGSNOR | LOC100499743 | TTGGAATGCTGCCACAAGGG | CCAGACACGTCCACTCACCA |
| GmNR | LOC100813471 | AACCGTCAATACGGCACCCA | TCGTCGTCGCTGGATGAGTC |
| GmNOX1 | LOC732612 | CAGAGCGCGGCTTTCACTTT | CACGTCATCGCTGTTGCTGT |
| ABAR | LOC100787735 | ATCAGGGCAATCAGATGGG | GAGGCGAAGACATTGAGATAGG |
| GmActin | LOC100797704 | GAGCTATGAATTGCCTGATGG | CGTTTCATGAATTCCAGTAGC |

15 min, denaturation at 95 °C for 15 s, and annealing and extension at 60 °C for 30 s. Actin was used as reference to evaluate the expression level of our genes of interest (Table 1).

## Statistical analysis

Each treatment was comprised of five pots, and each pot contained nine plants, thus making a total of 45 plants. All the biochemical and gene expression analysis was performed in triplicate. The recorded data were subjected to a two-way ANOVA with multiple comparisons and computation of CIs, and significance was determined using Sidak test with GraphPad Prism. The differences among the mean values were determined using Duncan's multiple range tests (DMRT) with significance at $P < 0.05$. The results were graphically presented using Graph Pad Prism software (version 5.0; San Diego, California USA), while Statistic Analysis System (SAS 9.1) was used for DMRT analysis.

# RESULTS

## Exogenous NO vs ROS antagonism during flooding stress

Short-term flooding-induced ROS generation, especially superoxide anion, is a common phenomenon. The results of superoxide anion accumulation displayed varying responses of plants to SNO and CySNO treatment and duration of flooding stress. Significantly low level of superoxide anions had accumulated in Daewon plants treated with SNP and CySNO before the flooding stress. We recorded 13% and 20% decrease in superoxide anions in these plants after 3 h and 6 h of flooding, compared to control flooded plants (Fig. 1A). A similar trend was observed in Pungsannamul cultivars that were subjected to flooding for 3 h, in which a 15% and 44% decrease in superoxide anion was recorded in plants treated with SNP and CySNO, respectively. Similarly, SNP and CySNO application also reduced the accumulation of superoxide anion by 11% and 46% after 6 h of flooding stress (Fig. 1B).

The glutathione-related antioxidant system is known to play a vital role in counteracting ROS generation. Our results demonstrated a significant increase in the accumulation of reduced glutathione content after flooding. However, plants treated with either of the SNO and CySNO as NO-donors showed a significant decrease in reduced glutathione content. For example, reduced glutathione contents were reduced by 14% and 21% in Daewon cultivars treated with SNP and CySNO, respectively, after 3 h of flooding (Fig. 2A). Continued exposure to flooding maintained a greater accumulation of reduced glutathione in Daewon as compared to the non-flooded control (Cont.) plants. However, in SNP and

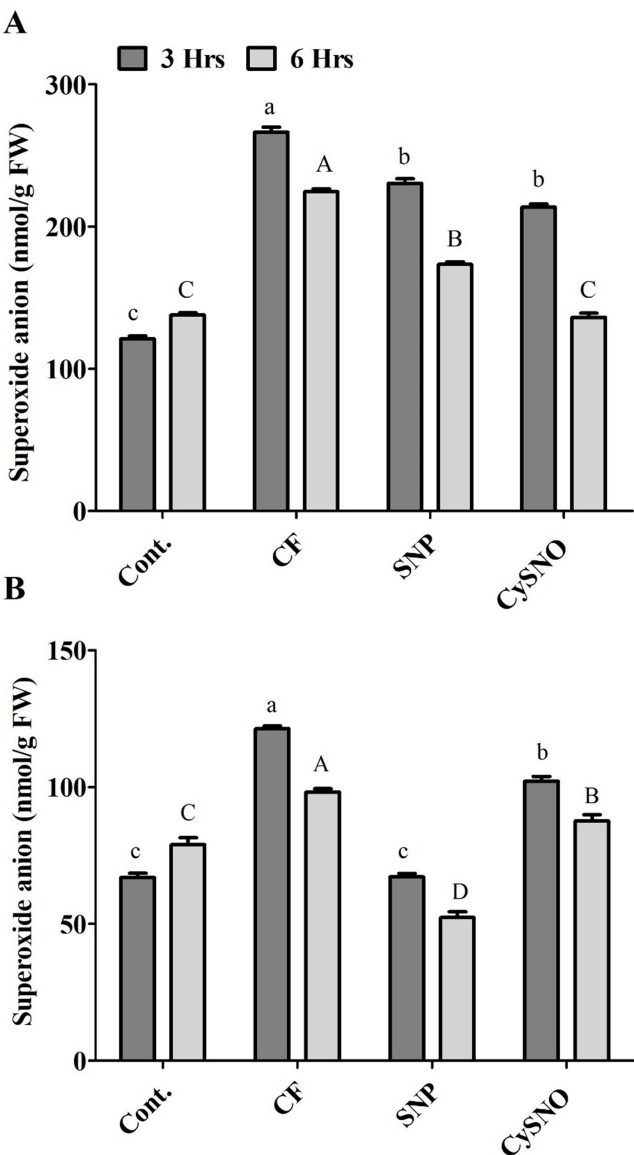

**Figure 1** **Application of exo-NO sources resulted in reduced ROS production, accumulation of superoxide anion in Daewon (A) and Pungsannamul (B) cultivars after 3 h (lowercase) and 6 h (uppercase) of flooding stress.** Abbreviation Cont. for control plants without any flood, CF for control with flood, SNP for Sodium nitroprusside application during flooding and CySNO for S-nitroso L-cysteine application during flooding. Data represent the mean of three replicates, while error bars represent standard errors. The differences among the mean values were determined using Duncan's multiple range tests (DMRT) at $P < 0.05$. The results were graphically presented using Graph Pad Prism software (version 5.0; San Diego, California USA), while Statistic Analysis System (SAS 9.1) was used for DMRT analysis.

CySNO treatments, a significant reduction in the reduced glutathione level (ranging from 16% to 24%) was observed in the Daewon cultivar after 6 h of flooding stress (Fig. 2A).

Flooding stress (3 h and 6 h) also caused a greater accumulation of reduced glutathione in the Pungsannamul soybean cultivar. The Pungsannamul soybean cultivar showed

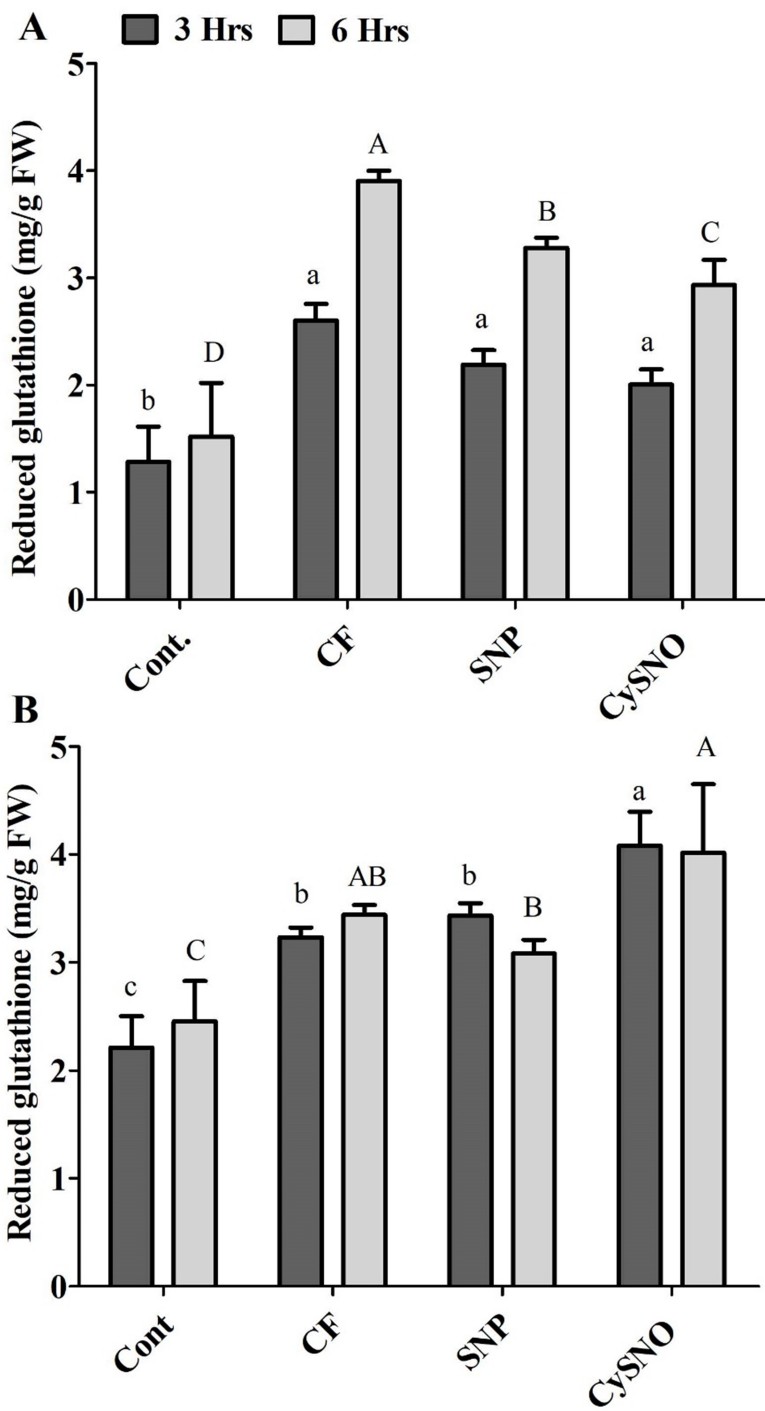

**Figure 2** **Reduced glutathione in exo-NO sources treated Daewon (A) and Pungsannamul (B) cultivars after 3 h (lowercase) and 6 h (uppercase) of flooding stress.** Abbreviation Cont. for control plants without any flood, CF for control with flood, SNP for Sodium nitroprusside application during flooding and CySNO for S-nitroso L-cysteine application during flooding. Data represent the mean of three replicates, while error bars represent standard errors. The differences among the mean values were determined using Duncan's multiple range tests (DMRT) at $P < 0.05$. The results were graphically presented using Graph Pad Prism software (version 5.0; San Diego, California USA), while Statistic Analysis System (SAS 9.1) was used for DMRT analysis.

significantly greater (23%) reduced glutathione accrual in SNP and CySNO treated plants, respectively, after 3 h of flooding stress (Fig. 2B). The continued exposure to flooding (6 h) influenced the Pungsannamul soybean plants to synthesize more reduced glutathione compared to non-flooding control plants and 3 h stress. The SNP treatment significantly reduced (10%) the synthesis of reduced glutathione after 6 h of flooding stress as compared to control flood treatments. CySNO, on the other hand, demonstrated a significantly larger (18.1%) content of reduced glutathione in the Pungsannamul cultivar (Fig. 2B).

## Regulatory networks involved in ABA synthesis during NO and flooding stress

The whole plant endogenous ABA synthesis was highly activated by flooding stress compared to normal growth conditions in Daewon plants. However, during the initial 3 h of flooding stress, whole ABA contents were significantly reduced (10% to 26%) by SNP and CySNO as compared to plants grown under flood conditions (Fig. 3A). However, with the continued exposure to flood stress (6 h), the endogenous whole ABA content sharply increased (63%) in the flooding control as compared to the non-flooding control. In this case, SNP and CySNO induced a significant decline (19% to 21%) in whole ABA biosynthesis (Fig. 3A).

ABA biosynthesis was comparatively less activated in the flood-sensitive Pungsannamul cultivar than in the Daewon cultivar. The whole ABA content was significantly higher (3 fold) in the control flooded plants than in non-flooding control plants. Compared to control, whole ABA contents in Pungsannamul were significantly reduced by SNP and CySNO application (11% and 14%, respectively). After 6 h of treatment, the whole ABA content sharply increased (63%) in the control flooded treatment, although SNP and CySNO-treated plants accumulated significantly less ABA (14%).

## Modulation of endogenous S-Nitrosothiols by exogenous NO donors under flooding stress

The endogenous S-Nitrosothiols (SNOs) were quantified in the shoots of both soybean cultivars. Compared to the control treatment, SNO accumulation reduced upto 58% in flooded plant however the SNO accumulated significantly in Daewon plants in response to CySNO and SNP treatments (23% and 90%, respectively) during 3 h of flooding stress compared to control flooded (Fig. 4A). After 6 h of flooding, SNP and CySNO application showed significantly reduced 7.66% and 10.61% endogenous SNO levels in Daewon cultivar (Fig. 4A).

Interestingly, endogenous SNO levels differed between Daewon and Pungsannamul cultivars during flooding, which could be attributed to the variation in their physiology and genetic response to flooding. In pungsannamul, compared to control treatment a significant higher accumulation of SNO level were observed in flooded plant (233.21%) however in control flooded, SNP and CySNO the endogenous SNO levels were detected in moderate to low ranges during 3 h of flooding stress (Fig. 4B). In SNP and CySNO treatments the SNO level were significantly increased from 30.81 to 53.84%. After 6 h of flooding stress, significantly higher SNO content (67%–140%) had accumulated in the

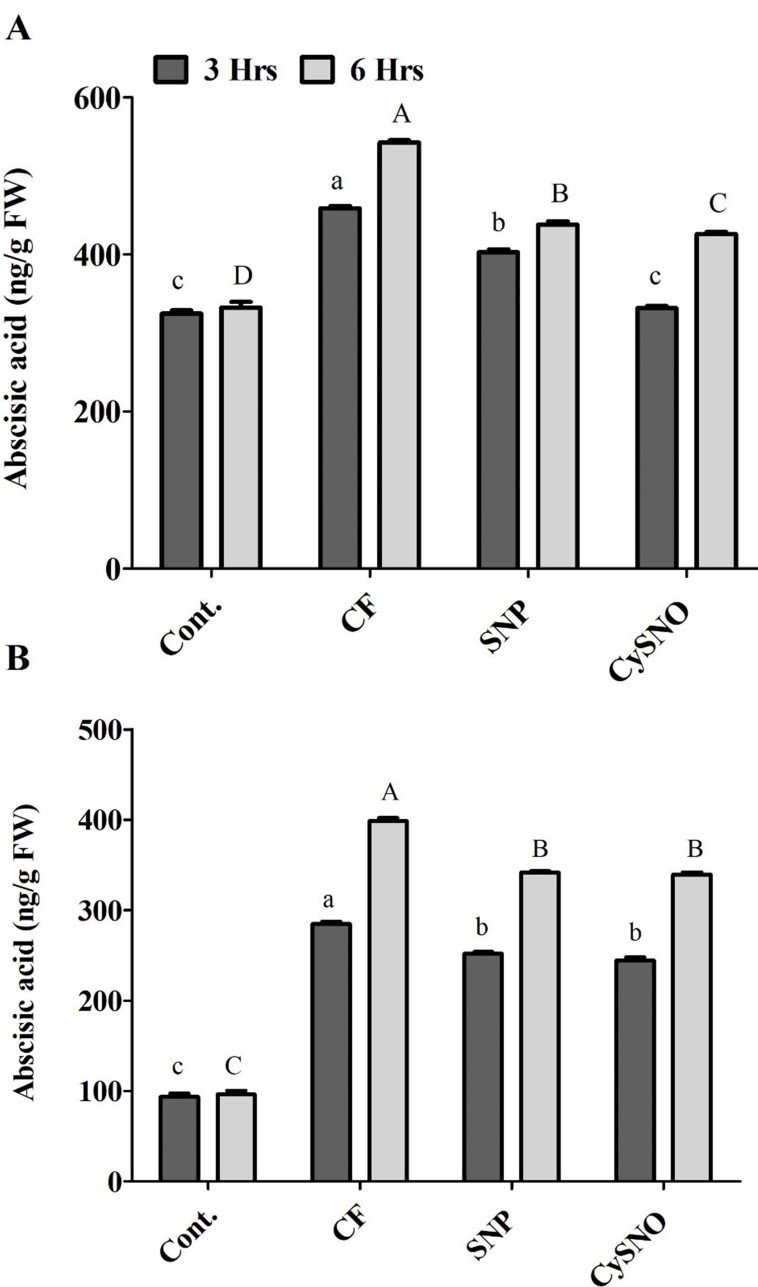

**Figure 3** **NO regulates cellular ABA levels in Daewon (A) and Pungsannamul (B) cultivars.** Total cellular ABA levels after 3 h (lowercase) and 6 h (uppercase) of flooding stress after application of NO sources. Abbreviation Cont. for control plants without any flood, CF for control with flood, SNP for Sodium nitroprusside application during flooding and CySNO for S-nitroso L-cysteine application during flooding. Data represent the mean of three replicates, while error bars represent standard errors. The differences among the mean values were determined using Duncan's multiple range tests (DMRT) at $P < 0.05$. The results were graphically presented using Graph Pad Prism software (version 5.0; San Diego, California USA), while Statistic Analysis System (SAS 9.1) was used for DMRT analysis.

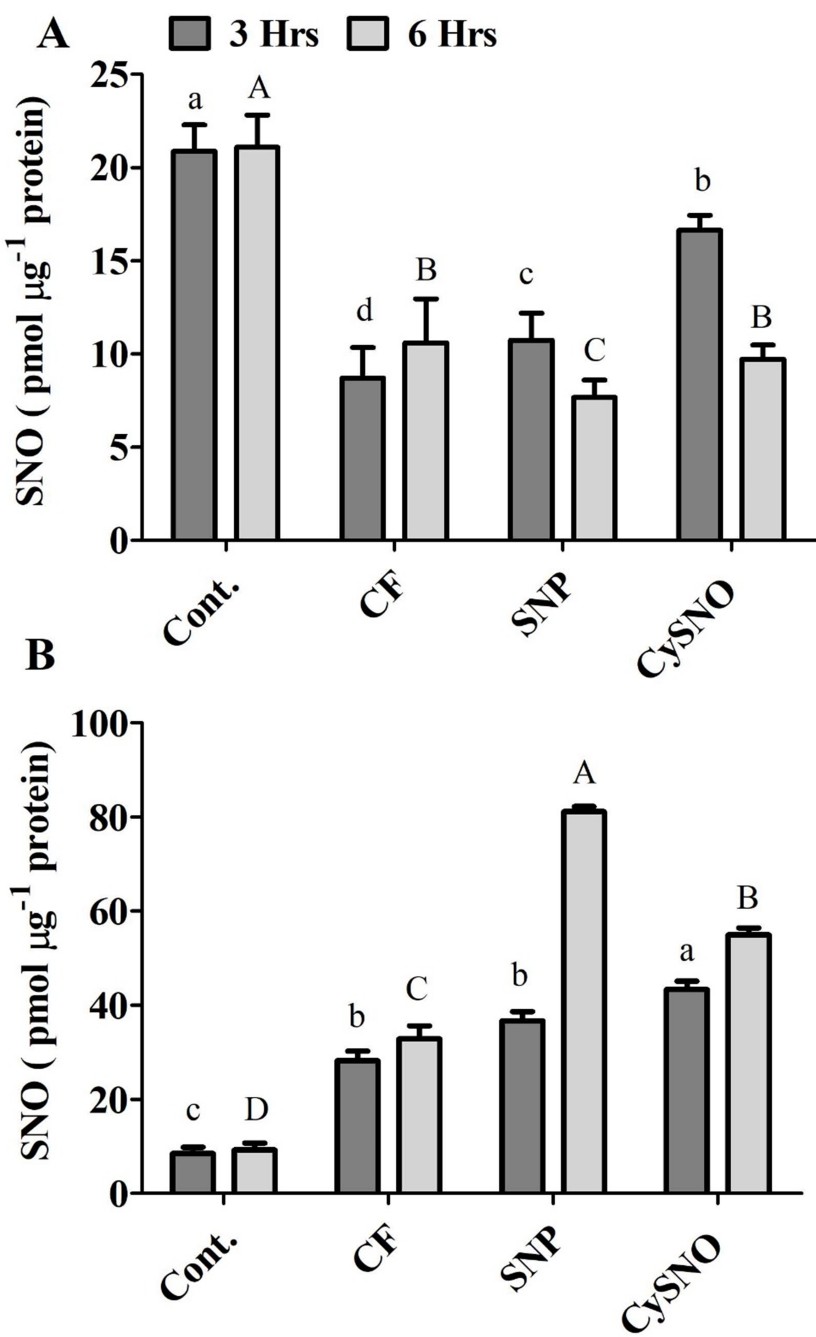

**Figure 4** **Cellular SNO levels of exo-NO sources treated Daewon (A) and Pungsannamul (B) cultivars after 3 h (lowercase) and 6 h (uppercase) of flooding stress.** Abbreviation Cont. for control plants without any flood, CF for control with flood, SNP for Sodium nitroprusside application during flooding and CySNO for S-nitroso L-cysteine application during flooding. Data represent the mean of three replicates, while error bars represent standard errors. The differences among the mean values were determined using Duncan's multiple range tests (DMRT) at $P < 0.05$. The results were graphically presented using Graph Pad Prism software (version 5.0; San Diego, California USA), while Statistic Analysis System (SAS 9.1) was used for DMRT analysis.

aerial parts of Pungsannamul in response to SNP and CySNO treatments as compared to the control flooding (Fig. 4B).

## Molecular analogy of exogenously applied-NO with SNO-related gene expression during flooding stress

Cellular SNO levels are controlled by a key enzyme *S*-nitrosoglutathione reductase (*GSNOR*) (*Feechan et al., 2005*). To observe whether flooding-induced SNO levels are regulated by *GSNOR*, we examined transcript accumulation of soybean *GSNOR1* (AT5G43940) in plants of both soybean cultivars after flooding stress alone and in those treated with SNO and CySNO as NO donors. The Daewon cultivar showed a lower expression of *GSNOR1* after 3 h of flooding, except in the CySNO treatment in which a significantly greater (2.1 fold) transcript accumulation of *GSNOR1* was observed. However, after 6 h of flooding, the *GNSOR1* transcript accumulation was significantly increased (2.9 fold) by SNP treatment as compared to the CySNO and control treatments (Fig. 5A). During the SNP and CySNO treatments, the Pungsannamul cultivar demonstrated a significantly increased transcript accumulation of *GSNOR1* (1.2 to 2.2 fold, respectively) as compared to the control after 3 h of flooding stress. This suggests a positive regulation of *GSNOR1* in modulating SNO levels under flooding stress. In contrast, there was no significant difference in *GSNOR* transcript accumulation at 6 h post flooding stress (Fig. 5B).

To further understand the possible route of flooding-induced NO production, we examined the transcript accumulation of *NOX1* (AT5G33320) and *NR* (AT1G37130). In the Daewon cultivar, the expression of *NOX1* was reduced overtime in response to both SNP- and CySNO-treated plants as compared to the control (Fig. 5C). However, in the Pungsannamul cultivar, SNP and CySNO treatments induced significantly increased (0.6 to 1.8 fold) levels of the *NOX1* transcripts compared to the control after 3 h of flooding (Fig. 5D). Conversely, these transcript levels were significantly obscured after 6 h of flooding.

In the case of *NR*, at 6 h post-flooding stress SNP and CySNO treatments showed either reduced or similar expression levels in comparison with the non-flooded and flooded controls (Fig. 5E). *NR* activation was predominant in the non-flooded Pungsannamul control as compared to CySNO and SNP treatments. *NR* transcripts were significantly higher in the SNP treatment compared to CySNO treatments during 3 h and 6 h of flooding periods but were significantly lower in the non-flooded plants (Fig. 5F). Additionally, the CySNO treatment demonstrated a gradual increase in *NR* transcript accumulation in both cultivars (Figs. 5E and 5F), whereas the opposite trend was observed for *GSNOR*.

## ABA-related transcript accumulation in response to flooding stress

One of the key consequences of water stress is the re-distribution and synthesis of ABA that regulates stomatal movements. NO is reported to be involved in ABA-mediated stomatal changes (*Garcia-Mata & Lamattina, 2002*). In this regard, we examined the transcript accumulation of *Timing of CAB expression 1* (*TOC1*) and *ABA-receptor* (*ABAR*) to understand NO-induced transcriptional changes during flooding stress. Our results showed reduced transcript accumulation of *ABAR* compared to control plants in the

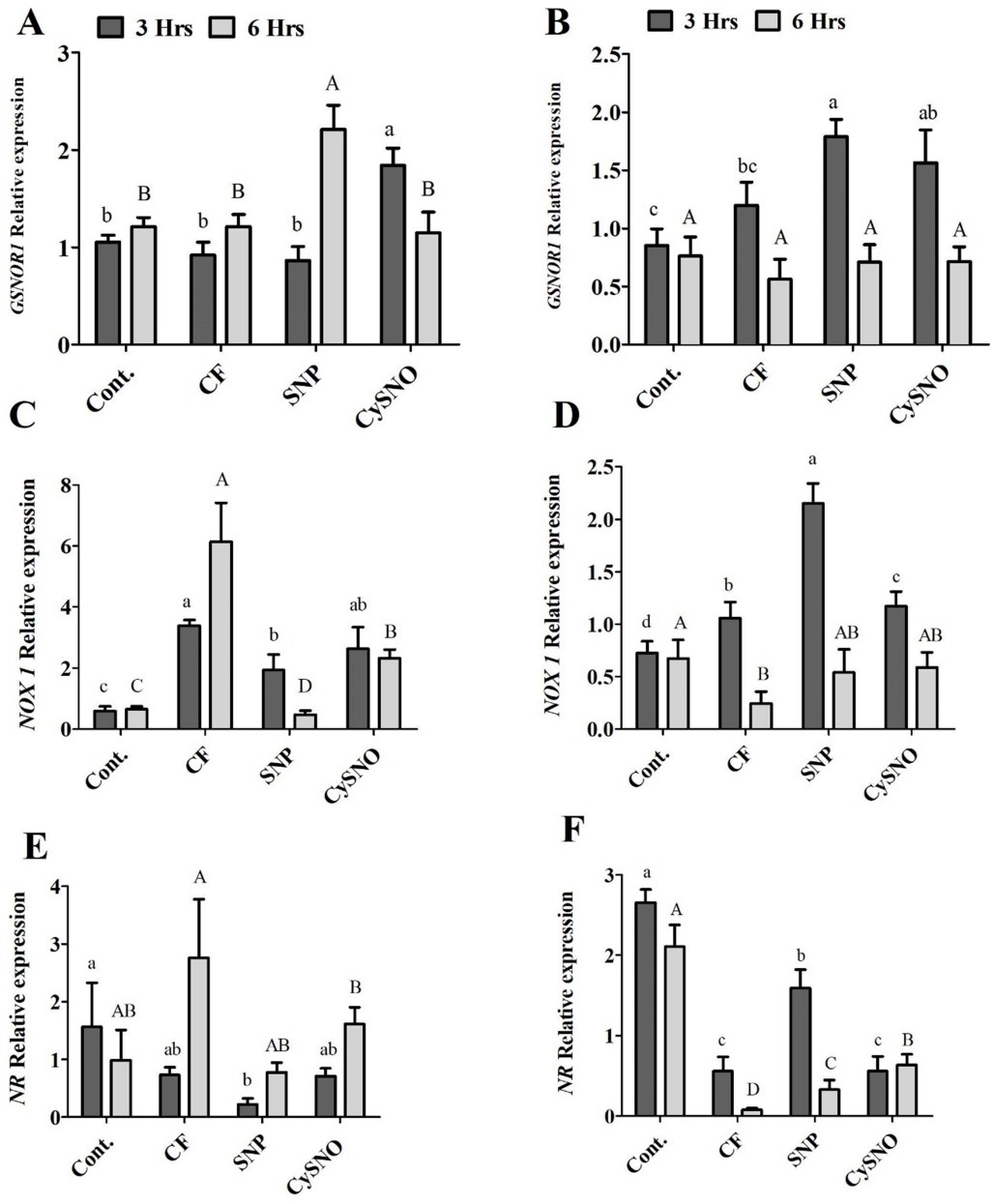

**Figure 5  Transcript accumulation of genes involved in nitrogen metabolism.** Expression level of *GSNOR*, *NOX1* and *NR* in exo-NO sources treated Daewon (A, C and E) and Pungsannamul (B, D and E) cultivars after 3 h (lowercase) and 6 h (uppercase) of flooding stress. The values were calculated relative to those of Actin gene and are the means of three replicates, while error bars represent standard errors. The differences among the mean values were determined using Duncan's multiple range tests (DMRT) at $P < 0.05$. The results were graphically presented using Graph Pad Prism software (version 5.0; San Diego, CA, USA), while Statistic Analysis System (SAS 9.1) was used for DMRT analysis.

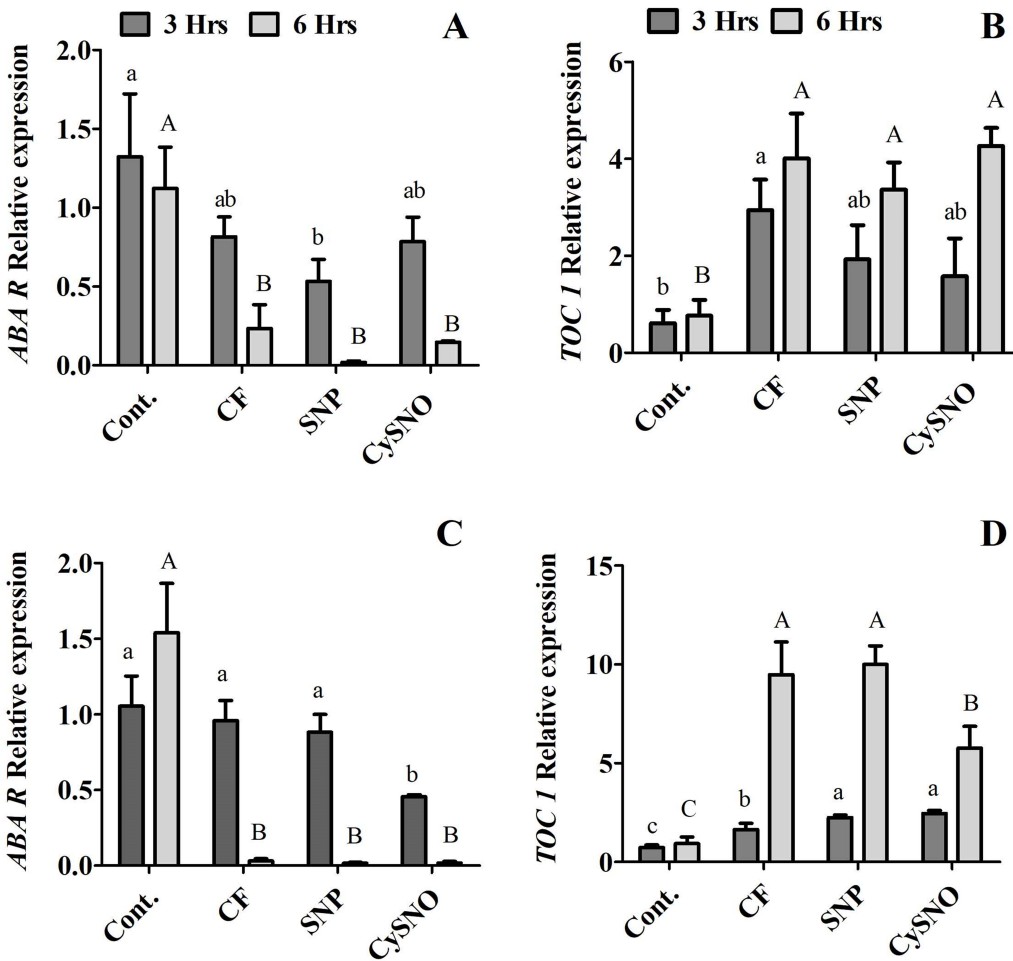

**Figure 6** **Relative expression of ABA synthesis genes.** *ABAR* and *TOC1* in exo-NO sources treated Daewon (A and B) and Pungsannamul (C and D) cultivars after 3 h (lowercase) and 6 h (uppercase) of flooding stress. Abbreviation Cont. for control plants without any flood, CF for control with flood, SNP for Sodium nitroprusside application during flooding, and CySNO for *S*-nitroso L-cysteine application during flooding. The values were calculated relative to those of Actin gene and are the means of three replicates, while error bars represent standard errors. The differences among the mean values were determined using Duncan's multiple range tests (DMRT) at *P* < 0.05. The results were graphically presented using Graph Pad Prism software (version 5.0; San Diego, CA, USA), while Statistic Analysis System (SAS 9.1) was used for DMRT analysis.

Daewon soybean cultivar (Fig. 6A). Similarly, *TOC1* showed a decrease (1.2 fold) in transcript accumulation in Daewon soybean plants treated with exogenous SNP after 6 h of flooding stress compared to 3 h flooding stress and the non-flooded control plants (Fig. 6B).

We additionally studied the expression level of *TOC1* and *ABAR* in the Pungsannamul soybean cultivar and observed no significant difference in *ABAR* expression level at 3 h of flooding stress. However, reduced transcript accumulation was observed for all treatments at 6 h of flooding compared to the non-flooding control plants (Fig. 6C). The transcript accumulation of *TOC1* showed that at 3 h of flooding stress the expression of *TOC1* was

induced only in plants treated with exogenous SNP and CySNO compared to control plants. In contrast, the transcript accumulation of *TOC1* was significantly induced at 6 h of flooding in both NO treated plants compared to the control plants (Fig. 6D).

## DISCUSSION

Recent studies have demonstrated that exogenous application of SNP and CySNO induces abiotic stress tolerance traits in plants, typically due to the activation of antioxidant metabolites or enzymes such as glutathione's (*Esringu et al., 2016*; *Jian et al., 2016*). In the present study, the exo- SNP and CySNO as NO donor displayed a similar trend of glutathione synthesis during 3 h and 6 h of flooding stress in Daewon plants, while the opposite was observed in the Pungsannamul cultivar. In the Daewon cultivar, glutathione synthesis increased during 3 h and 6 h of flooding but application of SNP and CySNO reduced glutathione content suggesting an adoption of defensive strategy to overcome oxidative stress. In contrast, Pungsannamul plants responded to oxidative stress by increasing glutathione synthesis, except in the SNP treatment. These results demonstrate the importance of SNP in the modulation of stress-aversion strategies. This was further validated by the results of superoxide anion content, as reduced oxidants were observed in SNP and CySNO treatment after 3 h and 6 h of flooding stress. For instance, SNP application was shown to reduce the oxidative stress in some plants through the regulation of antioxidants (*Esringu et al., 2016*; *Innocenti et al., 2007*; *Jian et al., 2016*). Glutathione, on the other hand, is a known determinant of cellular redox homeostasis (*Cheng et al., 2015*) and often increases with the level and intensity of stress (*Noctor et al., 2012*). In the current study, glutathione increased with the inception of hypoxic conditions but later decreased with exo-CySNO regulators in Daewon soybean, suggesting an improved defensive mechanism by both exogenous and endogenous factors. A reduction of glutathione post-anoxia has also been suggested by *Arbona et al. (2017)*.

Endogenous ABA, on the other hand, is known to be involved in signalling cascades related to plant defence and abiotic stress modulation in various crop plants including soybean (*Yin et al., 2017*). The ABA signalling pathway is composed of several elements including receptors, transcription factors, secondary messengers such as $Ca^{2+}$, ion transporters, hydrogen peroxide, and nitric oxide (*Zhao et al., 2016*). ABA tends to increase during flooding stress and was reported in Barbeton daisy, citrus plants, alfalfa, tobacco, pea, tomato, and apple (*Arbona & Gómez-Cadenas, 2008*; *Bai et al., 2011*; *Castonguay, Nadeau & Simard, 1993*; *Else et al., 1995*; *Hurng et al., 1994*; *Jackson, Young & Hall, 1988*; *Olivella et al., 2000*) under flooding stress. The ABA response to flooding may differ and depend on the plant species and duration of flooding. Other studies reported that flooding stress have associated ABA accumulation with an increase in reactive oxygen species (ROS), in *Glycine max* L roots (*Vantoai & Bolles, 1991*), *Zea mays* L. leaves (*Yan et al., 1996*), and *Triticum aestivum* L. roots (*Biemelt et al., 2000*). The results of the present study showed that NO sources (SNP and CySNO) significantly decreased ABA accumulation in the Daewon and Pungsannamul soybean. These results are also consistent with the transcript analysis of TOC1 and ABAR genes during flooding stress. Initially identified in Arabidopsis,

*Legnaioli, Cuevas & Mas (2009)* suggested that TOC1 binds to the promoter of ABA-related gene (thus called ABAR). Interestingly this binding is regulated by ABA or in other words ABA induction of TOC1 helps in binding with ABAR. *Legnaioli, Cuevas & Mas (2009)* also confirmed this through functional genomics study and found that ABAR knocked down lines were not able to induced TOC1 to bind with ABAR and further suggested both ABAR and TOC1 role in drought regulation. Later their orthologs in Soybean were found to mediate flooding and drought responses in soybean as well *Syed et al. (2015)*. Both genes are related to the circadian expression of abiotic stress phases, during which both genes exhibit a reciprocal regulation during stress conditions. Our results revealed that, upon exo-NO application, the ABAR was up-regulated in the initial 3 h flooding but showed significant down regulation after 6 h post-flooding. Similarly, differential response was observed for both cultivars in terms of TOC1 expression after flooding stress. TOC1 showed decreased transcript accumulation in Daewon while increased in Pungsannamul cultivar after 3 h of flooding stress. TOC1 is acutely induced by ABA, and this induction advances the phase of TOC1 binding and modulates ABAR circadian expression (*Legnaioli, Cuevas & Mas, 2009*); *Syed et al. (2015)*. This suggests that exo-SNP can regulate soybean plant responses by influencing ABA and possibly regulating the transcription of ABAR and TOC1 during short-term flooding stress (*Arbona et al., 2017*).

Endogenous SNP and CySNO production has been recently reported as a major regulator in plant signalling in response to potential stress conditions (*Correa-Aragunde, Graziano & Lamattina, 2004*; *Neill et al., 2008*; *Yun et al., 2016*). A major route for the transfer of NO bioactivity is S-nitrosylation, the addition of an NO moiety to a protein cysteine thiol forming an S-nitrosothiol. Total cellular SNOs are controlled predominantly by S-nitrosoglutathione reductase 1 (GSNOR1) which turns over the natural NO donor, S-nitrosoglutathione (GSNO) (*Feechan et al., 2005*; *Malik et al., 2011*; *Yun et al., 2016*). In the absence of the GSNOR1 function, GSNO accumulates, leading to an overall rise in total cellular SNOs. The present results demonstrated an interesting response by increasing endogenous NO levels after 3 h relative to 6 h in Daewon soybean cultivar. In Pungsannamul soybean, higher level of endogenous NO was observed after 3 h and 6 h of flooding in shoots. These results suggest that exo-SNP and CysNO can mitigate the flooding stress-induced NO production with an exposure period in the Daewon cultivar, whereas in the Pungsannamul cultivar, the shoots would not able to counteract the stress. Nitric oxide transfer its bioactivity through post-translational modification called *S*-nitrosylation; the attachment of NO moiety with exposed cysteine thiol of other proteins to form S-nitrosothiol (SNOs) thereby regulating proteins functions. Therefore, the level of SNOs in cell may suggest the level of nitrosative stress and that is why its measurement is key parameter in number of studies involving role of NO in plants (*Feechan et al., 2005*; *Imran et al., 2016*; *Imran et al., 2018*; *Yun et al., 2011*). We therefore sought to determine SNO levels as reactive nitrogen species to assess the magnitude of nitrosative stress in NO-donors treated samples after flooding conditions.

Contrasting NO production in Daewon and Pungsannamul varieties were observed in the present study also indicates the effect of exo-SNP and CySNO on plant growth. We did not notice any differential changes in plant morphology due to short-term flooding stress,

although the endogenous NO levels significantly varied in Daewon and Pungsannamul after 3 h and 6 h post-flooding periods. At 3 h post flooding stress the accumulation of SNO in SNP and CySNO treated plants increased. However, at 6 h post stress the SNO contents decreased suggesting that no more SNP or CySNO released NO could be accommodated in the cell as there is enough SNOs accumulated. Furthermore, the differential accumulation of NO in response to SNP could also be attributed to the limitations in use of SNP as NO donor. Contrasting reports are published mentioning both pros and cons of SNP as NO donor. In a study using germination assays in response to SNP and cyanide *Da Silva et al. (2019)* reported that cyanide released from SNP can increase germination of *Senna macranthera*. On the other hand reports suggested that cyanide release from SNP can reduce the photochemical activity of photosystem II thereby affecting plant productivity (*Wodala, Ordog & Horvath, 2010*). However, the difference in the reduction of SNOs might be due to the different No release by different NO donors which is also described by *He & Frost (2016)*. *Hussain et al. (2016)* used it as NO donor to see global changes in gene expression and identified more than 6,000 genes that should significant differential expression. Similarly, almost similar treatment of CySNO was used in other studies as well *Lam et al. (2010)* and *Martinez-Ruiz & Lamas (2004)* etc. Increased NO levels in both cultivars may indicate the transduction of NO bioactivity in plants after short-term flooding stress.

NO being more reactive needs to be converted into a more stable form called SNO (*Feechan et al., 2005*). The enzyme GSNOR play role in maintaining this hemostasis (*Leterrier et al., 2011*). GSNOR1 predominantly regulates S-nitrosylation via trans-nitrosylation (*Foster et al., 2009*), which involves the direct transfer of NO from GSNO to its target cysteine (Cys) residue. In the current study, it was coupled with transcript accumulation and reduction of GSNOR1 in response to 3 h and 6 h post-flooding stress periods in Daewon soybean. However, in Pungsannamul soybean plants, the exogenous SNP and CySNO reduced the transcript accumulation of GSNOR1 with the exposure of flooding stress. In the case of NR and NOX, a similar reduction in transcript accumulation was observed in Pungsannamul soybean plants. In Daewon plants, however, a reverse transcript expression was shown for NR and NOX genes. These findings were unexpected, as in previous work the NOX mutants exhibited increased NO accumulation in Arabidopsis thaliana (*Guo, Okamoto & Crawford, 2003*), which had negatively regulated the examined morphological responses (*He et al., 2004*). This is also in line with the ABA results in Daewon soybean plants suggesting that NR, which determines the NO production in plants, is critical to ABA-induced stomatal closure (*Kwon et al., 2012*; *Yun et al., 2016*; *Zhao et al., 2016*). The Pungsannamul soybean plants expressed the lowest level of gene expression of GSNOR1, NR, and NOX, suggesting a contribution towards the loss of potential function against prolonged flooding stress, which was also confirmed from a previous study involving Arabidopsis thaliana. Although efforts have been made to understand the role of SNO during innate immunity in plants against disease resistance, little is known about the comprehensive regulations of SNO synthesis and transcript accumulation under flooding stress. Our study urges that more attention be given to the important inter-junctions related to the role of NO in stress signalling in plants.

## CONCLUSION

In current study, we observed that exo-SNP and CySNO application as NO donor reduces glutathione activity and endogenous ABA over time, whereas increases S-nitorosothiols levels. Exo SNP and CySNO reduced ROS production but enhanced superoxide anion accumulation. The transcript accumulation of *GSNOR1, NOX1*, and *NR* were found in higher amounts under flooding stress as compared to control. Current results demonstrate an active role of exo SNP and CySNO during short-exposure to flooding stress. The exogenous SNP and CySNO impinge a variety of biochemical and transcriptional changes in soybean. Such changes have been suggested important to reprogram the stress responses by plant to maintain a study growth potential during shorter or longer flooding excursion.

**Abbreviations**

| | |
|---|---|
| **NO** | Nitric oxide |
| **TOC** | Timing of CAB expression |
| **GSNOR1** | S-Nitroso Glutathione Reductase1 |
| **NR** | Nitrate Reductase |
| **SNP** | Sodium Nitroprusside |
| **CySNO** | S-Nitroso L-Cysteine |
| **ABAR** | ABA-receptor |
| **ABA** | Abscisic acid |
| **GA** | Gibberellin |
| **SNO** | S-nitrosothiol |

### Funding

This research was supported by the Kyungpook National University Research Fund, 2018. The funders had no role in study design, data collection and analysis, decision to publish, or preparation of the manuscript.

### Grant Disclosures

The following grant information was disclosed by the authors:
Kyungpook National University Research Fund.

### Competing Interests

The authors declare there are no competing interests.

### Author Contributions

- Muhammad Aaqil Khan and Sajjad Asaf performed the experiments, prepared figures and/or tables.
- Abdul Latif Khan conceived and designed the experiments, analyzed the data, authored or reviewed drafts of the paper.
- Qari Muhammad Imran performed the experiments.
- Sang-Uk Lee performed the experiments.

- Byung-Wook Yun conceived and designed the experiments, contributed reagents/materials/analysis tools.
- Muhammad Hamayun analyzed the data, authored or reviewed drafts of the paper.
- Tae-Han Kim contributed reagents/materials/analysis tools, approved the final draft.
- In-Jung Lee conceived and designed the experiments, approved the final draft.

## Data Availability

  The raw measurements are available in the Supplemental Files.

## Supplemental Information

Supplemental information for this article can be found online at http://dx.doi.org/10.7717/peerj.7741#supplemental-information.

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
