# Peer review of "Exogenous application of nitric oxide donors regulates short-term flooding stress in soybean"

_PeerJ, doi:10.7717/peerj.7741_

## Round 0.1 · original submission · Major Revisions

Please improve the description of your experimental strategy and methods.

The reviewers also point towards several reports that should be taken into account.

I also recommend you to add some graphical material that illustrates the study/ overall experimental design.

Reviewer 1 ·

Basic reporting

1. Clear and unambiguous, professional English used throughout.
2. Introduction needs simplification.
3. Line 58, 59 “Over time, submergence……plant death” and “High ethylene ……root elongation” is lacking in excess.
4. Literature references, sufficient field background/context provided. But please modify the format of the references in the text, for example, line 46,55, 129, 131, 355, 444 , and check and revise the references section of the article.
5. What does the different uppercase and lowercase letters represent in Figures? Please add.

Experimental design

6. The title of 2. 1. does not correspond to the content, and this section does not describe the flooding stress.
7. Please add water, fertilizer, light, temperature and humidity management to the seedlings planted behind plastic pallets in part 2.1.
8. On line 154- what is stage VI? On line 159 what is stage 3 (V3)?

Validity of the findings

9. No references should be included in the results .
10. Line 450 "Recently, very little information are available on the ameliorative role of NO under flooding stress" has nothing to do with this study and is suggested to be deleted.

Reviewer 2 ·

Basic reporting

In this manuscript, Khan et al., report the effects of NO donors on various parameters (SNO groups, gluthatione activity, superoxide, ABA content, mRNA transcripts of NO associated genes) in waterlogged soybean cultivars. While it is clear that NO plays an important role in abiotic stress signaling and more robust data on NO dynamics in flooded plants is welcome, it is partly because NO applications and dynamics are hard to study and interpret. While I am happy with the readability of the manuscript and the authors attempting to contribute data to the field, I have several issues with the current manuscript with regards to reporting and experimental design that in my opinion require clarification. I will also raise some minor comments.

English writing/grammar: Report is relatively clearly written and structured but should be improved on English grammar/vocabulary. Some English could be improved such as in the first sentence of the introduction (editor or English native/expert should go through the manuscript). 45: FloodING is…. And the first sentence of the conclusion: 450: Recently, very little information IS… (and it is not recently that little information is available).
Also, punctuation and lack of use of capitols is sometimes wrong. I.e. 101: Therefore, etc..

Intro and background: Flooding and soybean losses are well reported and are adequately referenced. Also the overview of NO production and interactions with various pathways are well described. However I have issues with the one sided use of NO as a signaling molecule that is solely reported to be beneficial for abiotic stress tolerance. There are many studies described where NO actually is negatively correlated with survival for instance in the context of the N-end rule pathway or phytoglobins (Gibbs et al., 2014, Hebelstrup et al., 2012, overview in Sasidharan et al., 2018). This should be addressed in the introduction. Additionally, the application of NO (donors) is not necessarily adaptive itself, but induces pathways such as ethylene or cyanide induced signaling which are directly adaptive (Mira et al,. 2016).

Figures are described and labeled adequately, although sample size should be clarified. In the raw data, it appears only 3 replicates were used for all experiments. But from the methods sections these appear to be the means from each separate experiment consisting of 5 replicates (itself containing 9 pseudo-replicates?). Not really clear at first view. Raw data is only the mean processed values of each experiment. Aren't these relative (for instance for RT-qPCR results, raw CTs, technical replicates, housekeeping gene results) How were these calculated? What are the units?

Experimental design

The biggest issue I have with the r is the experimental design that is not clear enough to me to interpret the results (and the meaning of these results).

First of all, the soybean plants are waterlogged in either normal water or supplied with an NO donor. The authors should describe how this NO donor is applied, is it in the water that is used for waterlogging? When? At the start or at the 3h and 6h time-points? SNP and SNAP donate NO GAS (only in the light) and this will thus penetrate and affect only above water tissues (gas diffusion underwater is reduced 10.000 fold). In addition, SNP and SNAP action decrease rapidly in the light and produce important side effects such as cyanide production . Sole cyanide application effects have mimicked the effects of SNP as an NO donor, thus attributing the effects to KCN application and not NO (Wodala et al., 2010). Therefore, the application should be very well described and the results should be very carefully interpreted and in this instance can’t necessarily be attributed to NO alone.

Secondly, since waterlogging is used as a treatment, what are the harvested tissues? Leaves, whole shoots? Of the above water or submerged parts of the plant? This is very important to clarify, as the submerged parts will have a drastically altered physiological state compared to the above water parts.
Submergence is associated with hypoxia stress, but this will not be the case for the above water tissues (see any plant flooding review), and possibly not even for the below photosynthetic parts in the light. The tissues affected possibly by hypoxia, ethylene, ROS/NO, CO2 accumulation are the submerged parts, but the treated parts with NO donors are predominantly the above water parts (see first issue). In my opinion, the authors should state very clearly what and why this experimental design was used before anyone can draw any conclusions on the obtained results.

Finally, what are the positive controls to show that the NO donor treatments actually increased NO levels? If it is not possible to measure NO levels directly, could an NO scavenger be used simultaneously to attribute the discovered results to actual NO effects, and no for instance KCN byproducts?

This does not mean that the obtained results are not useful for the research field, but should be placed into context of the experimental design which is impossible for any critical reader right now. I would gladly see any revised version where the experimental design is clarified.

Validity of the findings

See above, as the validity of the findings is related to experimental design.

Additional comments

-

·

Basic reporting

This paper needs careful editing throughout.
I think there is quite a lot of background about NO and flooding missing. It is not quite the virgin territory that the authors suggest. See Dordas et al., 2003 for example. I think it is fair to say here that hypoxic stress is relevant and therefore there is a significant body of literature which could be included that involves NO (eg Wany 2017) although I realise they are not writting a review article here.
The article is structured appropriately and is self-contained with relevant data.

Experimental design

Although the experimental design is generally overall okay I struggled with the methods section which I think needs far more information in places and some information removed. How to make standard solutions can be taken out but I didn't understand the method on ROS generation, for example. This of course then impinges on the interpretation of the data. Literature citations need to be added to some of the methods too. Centrifugation should not be quoted as RPM but as xg. Then the section on SNO is primarily on protein quatification which could be a separate section and the SNO section done properly. Overall, the methods all need careful re-writing for clarity.
Can the authors say why NO was not measured, as this would have added to the robustness of the data?
No SNP and CySNO depleted controls were used and these ought to be done. SNP has by-products which are bio-active, such as cyanide (Bethke's work). Therefore controls are important. No NO scavengers were used either.
There are no ethical considerations needed.

Validity of the findings

The presentation and description of the data is fine. The data has statistical analysis and it is clear to see the data. However, please revisit figure legends and make it clear what panels A,B,C etc are. It is not always stated and one has to read the text section to find out or guess.
The conclusions are quite long and rambling and tend to repeat the results. In places key issues are skimmed over so the whole section could be re-visited and written in a tighter manner.

Additional comments

This is an interesting study but methods need to be clearer, and the background and discussion more focused and inclusive of flooding/hypoxia/NO literature.
Lastly, i would be good to know why the plant varieties were chosen, whether the authors expected the differences and discuss why they might be different, as the differences are both significant and interesting.

---

## Round 0.2 · Major Revisions

Both reviewers still have major concerns about the validity of your conclusions. In particular, the comments of the reviewers need to be addressed carefully and integrated in the revised manuscript.

[]

Reviewer 2 ·

Basic reporting

I am very happy to see that basic reporting has improved significantly in response to the reviewer comments, although I still find that English grammar and sentence structure should be addressed more seriously. Especially of the newly written parts.

- Right now, the statistical lettering in the figures is not clear to me. What do the minor and capitol letter represent, and do there represent statistically different groups? This should be clearly explained in ever single figure legend.

- i still think that raw data should include more than the processed values. For instance, of the qPCR, also the raw CTs of the genes of interest and the housekeeping genes.

Experimental design

The authors attempted to address my concerns in their rebuttal, but did not incorporate their responses in their manuscript. I find that they also did not address the technical issues raised by reviewer #3, in the manuscript. In other words, Non stressed NO controls are still not included, or a verification to detect whether NO application was successful. They claim that NO application could be verified by their SNO measurements, but these actually show that NO leads to lower SNO in one of the 2 cultivars, while one would expect higher SNO content if NO application was succesful. If the authors do not want to re-do experiments or include additional, but essential controls, they should discuss the limitations of SNP treatments somewhere (methods or discussion). For instance, indicate that SNP treated results could be attributed to cyanide, and not NO (Wodala et al., 2010).

To be specific this has to be explained in the manuscript to allow for reproducibility and validity:

- In my opinion, the potential flaws of using SNP as an NO donor should be addressed either in the methods but preferably the discussion. As the researcher can not rule out that the observed effects are not caused by actual NO effects, but other bio-active compounds released by SNP.

- Please clearly state which materials were harvested (whole shoot tissues, only leaf tissues?/etc). This is essential for reproducibility.

- Explain why no non-waterlogging SNP controls were included. If the SNP and CysNO treatments were succesful, would you not expect SNO content to go up in all of these treatments?

- The authors claim in the text (line 321) to have measured SNO in the roots as well, although I do not see these data anywhere. If they have this data, this would be very valuable to share as well, as these are the only tissues suffering from hypoxia.

Validity of the findings

See experimental design section. The data might be robust but have to be clearly placed into the context of the limitations of using SNP (cyanide in the water) as an NO donor and the lack of non-waterlogged NO treated controls.

·

Basic reporting

see below

Experimental design

see below, this is flawed as the paper stands.

Validity of the findings

see below, again flawed as paper stands.

Additional comments

The authors have considered the comments of the referees and the manuscript is better but I do have a major issue with it still. The paper, and I quote that: "we observed that exo-NO application reduces..."
BUT I don't agree. It may be the case but the experimental design and analysis does not support this. In the rebuttal they say: "our sole focus was to understand the effect of most common exo NO source (SNP & CySNO) ..." which is true. This paper reports on the effects of SNP and CySNO, but it cannot claim to report the effects of NO. There are simply not enough controls carried out. I suggested measuring NO and yes, they are adding NO donors but it would be nice to know that NO was increased. There are no scavengers or depleted controls used. So, I would be happy if the paper is re-titled and re-written to be clear that the effects seen are after treatment of SNP and CySNO, but not NO. Yes, this would limit the impact of the paper, but as it stands I think it would be misleading to continue to claim that the effects were from NO.

---

## Round 0.3 · accepted · Accept

We especially appreciate that you provide your raw data as supplemental material.